# Mycorrhizal types influence island biogeography of plants

Camille S. Delavaux [1✉], Patrick Weigelt[2], Wayne Dawson [3], Franz Essl[4], Mark van Kleunen [5], Christian König[2], Jan Pergl[6], Petr Pyšek[6], Anke Stein[5], Marten Winter [7], Amanda Taylor[2], Peggy A. Schultz[1], Robert J. Whittaker [8], Holger Kreft [2] & James D. Bever [1]

Plant colonization of islands may be limited by the availability of symbionts, particularly arbuscular mycorrhizal (AM) fungi, which have limited dispersal ability compared to ecto-mycorrhizal and ericoid (EEM) as well as orchid mycorrhizal (ORC) fungi. We tested for such differential island colonization within contemporary angiosperm floras worldwide. We found evidence that AM plants experience a stronger mycorrhizal filter than other mycorrhizal or non-mycorrhizal (NM) plant species, with decreased proportions of native AM plant species on islands relative to mainlands. This effect intensified with island isolation, particularly for non-endemic plant species. The proportion of endemic AM plant species increased with island isolation, consistent with diversification filling niches left open by the mycorrhizal filter. We further found evidence of humans overcoming the initial mycorrhizal filter. Naturalized floras showed higher proportions of AM plant species than native floras, a pattern that increased with increasing isolation and land-use intensity. This work provides evidence that mycorrhizal fungal symbionts shape plant colonization of islands and subsequent diversification.

[1] University of Kansas, Lawrence, USA. [2] University of Gottingen, Göttingen, Germany. [3] University of Durham, Durham, UK. [4] University Vienna, Vienna, Austria. [5] University of Konstanz, Konstanz, Germany. [6] Czech Academy of Sciences, Průhonice, Czech Republic. [7] German Centre for Integrative Biodiversity Research, Leipzig, Germany. [8] University of Oxford, Oxford, UK. ✉email: camille.delavaux@usys.ethz.ch

Classical island biogeography recognizes that species richness results from the net effects of immigration, extinction, and speciation. These biogeographical rates have been primarily linked to abiotic features of islands: immigration decreases with isolation, extinction decreases with island size[1], and speciation increases with island size and isolation[2,3]. However, only a limited number of case studies have addressed how biotic interactions influence colonization, extinction, and speciation probabilities on islands[4–6], and generalizations are difficult. Order of arrival, resulting in priority effects[7], is likely to be particularly important for mutualisms. The mycorrhizal mutualisms formed between soil fungi and most plant species are prime candidates for priority effects.

Many plant species are highly dependent on mycorrhizal fungi[8], however, these fungi disperse independently of their plant hosts and therefore the absence of mycorrhizal fungi may limit the colonization of mycorrhizal plant species[9]. Indeed, a recent global analysis of native floras found both a lower proportion of mycorrhizal plant species on islands than on mainlands and a decrease in the proportion of mycorrhizal plant species in island floras with increasing isolation[10], consistent with the operation of a mycorrhizal filter on plant colonization of islands. However, whether different mycorrhizal fungal types differentially impact the composition of island floras is currently unknown. Here, we test for differences in the strength of the mycorrhizal filter across plant species associating with the three major mycorrhizal types: arbuscular mycorrhizal (AM) fungi, the most common type of mycorrhizae, ectomycorrhizal and ericoid mycorrhizal (EEM) fungi, and orchid mycorrhizal (ORC) fungi.

We can construct two contrasting a priori sets of expectations for the relative strength of the mycorrhizal filter based on differences in the biology of different types of mycorrhizal fungi. AM fungi are likely to be most limited in their ability to colonize islands prior to host plants due to two life-history traits. First, AM fungi lack adaptations for aerial dispersal (but see Chaudhary et al.[11]). Moreover, AM fungi cannot grow independently of their hosts[12]. In contrast, individual fungal species of other types of mycorrhizae, particularly EEM, have adaptations for aerial dispersal of spores[13,14] and can grow independently of their host through saprophytic activity[12,15–20] (but see Lindahl et al.[21]). We therefore expect EEM and ORC fungi to be better able to establish on islands prior to their hosts compared to AM fungi, and EEM and orchid plant species to be less impacted by the mycorrhizal filter than AM plant species[22].

Plant colonization success could also be impacted by specificity within these associations. Mycorrhizal associations with low fungal-plant specificity are less likely to limit plant establishment because the establishment of a single fungal species could enable the colonization of many plant species[23]. Alternatively, in associations with high specificity, the establishment of a single fungal species may only enable colonization of a small subset of the plant species of that mycorrhizal type. AM fungi have lower specificity of association than EM and ORC[12,24,25], thereby reversing expectations for the strength of the mycorrhizal filter from those based on colonization ability. Finally, the extent to which plants are obligately dependent on mycorrhizal fungi could modify the potential for these fungi to limit plant colonization of islands, with facultatively dependent plants colonizing islands more easily. A greater proportion of plants that associate with AM than EM fungi have been identified as facultatively dependent on mycorrhizal fungi[26]. This would again generate patterns counter to dispersal expectations but consistent with specificity expectations, where AM plants experience a weaker filter than EEM or orchid plant species.

Besides acting as a filter on colonization, the types of mycorrhizal associations may influence the global distribution patterns of plant species through functional differences, providing additional hypotheses relevant to global biogeography. For instance, AM fungi are thought to be most effective at facilitating access to relatively immobile resources such as inorganic phosphorus and nitrogen released by saprotrophs, and EM fungi are commonly thought to be able to better access organic nitrogen[27], potentially short-circuiting the decomposition pathway. This function is assumed to be particularly important in colder climates where decomposition is slow. These differences underlie arguments for the dominance of EM plant species in colder climates[16]. Recent analyses built on assumptions of the ecological differences in AM and EM symbioses predict extant patterns of mycorrhizal types in forests, with greater dominance of AM plant species near the equator and greater dominance of EM plant species closer to the poles[28,29]. Therefore, we expect the biogeography of AM and EEM mycorrhizal plant species to be driven in part by temperature and precipitation, important decomposition-related environmental variables that vary with latitudes. Predictions based on functional differences of ORC fungi are difficult, as the associated plants can be parasitic rather than mutualistic with their fungi[30,31].

Here, we explore biogeographical patterns of angiosperm species that associate with different types of mycorrhizal fungi. We use a global database to test for persistent legacies of the differential strength of a mycorrhizal filter in island colonization as predicted by differences in dispersal-dependence and host specificity of these fungal groups. We test for differences in the proportion of plant species that associate with different types of mycorrhizal fungi between mainland and island systems in both native and naturalized floras. We then examine endemism patterns in native island floras to confirm that colonization patterns, independent of diversification, are consistent with our analyses. A priori, we expect that plant types most affected by the mycorrhizal filter will have higher diversification rates to fill niches left open by limited colonization, and therefore higher rates of endemism. Finally, we analyze the potential drivers of these patterns by predicting these proportions based on geographical, environmental, and anthropogenic variables.

We demonstrate that, worldwide, AM plant species show colonization limitation on islands, and this effect increases with distance from the nearest mainland. We further find that endemism increases most with distance for AM species, consistent with greater unfilled niche space on these remote islands. When examining the naturalized flora, we find that all mycorrhizal types are overrepresented, confirming that these mycorrhizal species are a higher invasion risk for islands. Overall, we show that the AM symbiosis limits AM plant species' establishment on islands more than other mycorrhizal associations, and that this initial filter of AM plant species impacts diversification and plant invasion risks.

## Results

**Evidence of differential mycorrhizal filters in native oceanic island floras.** Across oceanic islands globally, we found support for dispersal limitation of native angiosperm plant species that associate with AM fungi (Figs. 1a and 2a, c, supplementary tables 1–6). Specifically, compared to mainland floras, we found that native island floras had ~5% lower proportion of AM than EM (EEM:AM $p < 0.001$; Supplementary Table 2a M1) and 10% lower AM than NM plant species (AM:NM $p < 0.001$; Supplementary Table 4a M1). Moreover, the proportion of plant species on islands that associate with AM relative to NM significantly decreased with increasing distance from the mainland ($p < 0.01$, Supplementary Table 4a M4, Fig. 3a, and Supplementary Fig. 1). When examining the proportion of endemic plant species, we

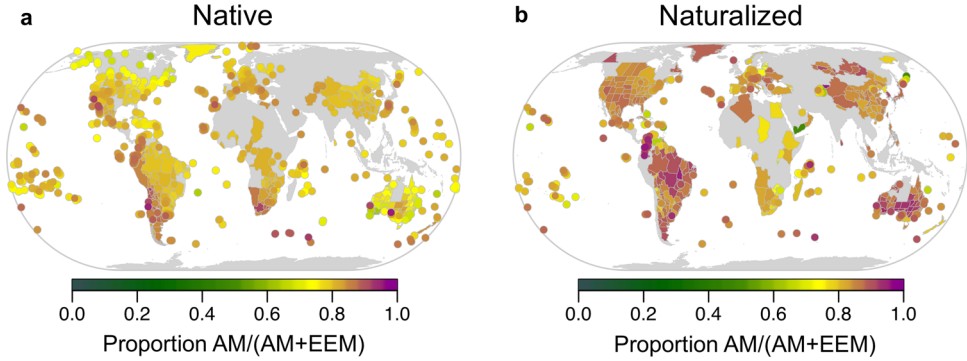

**Fig. 1 Locations of regions used in study with associated proportion AM:EEM.** Maps of geographical regions showing the proportion of arbuscular mycorrhizal relative to ectomycorrhizal (AM/AM + EEM) plant species for native and naturalized floras included in this study (**a** mainland $n = 515$, island $n = 313$; **b** mainland $n = 287$, island $n = 100$).

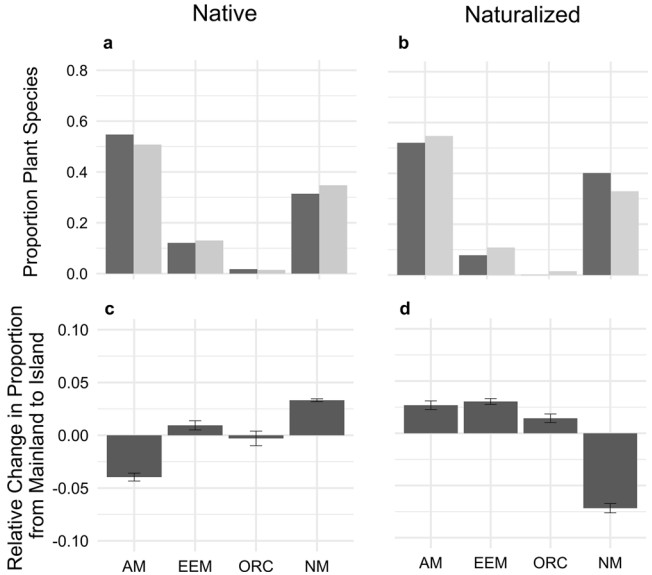

**Fig. 2 Proportion of plant species of each mycorrhizal type in native and naturalized flora.** Arbuscular mycorrhizal (AM) plant species represent a lower proportion of species in the native floras on oceanic islands (light gray) than on mainlands (dark gray), while all mycorrhizal types represent a higher proportion of species on islands than mainlands in the naturalized flora. Proportion of plant species within each mycorrhizal type: arbuscular mycorrhizal (AM), ecto- and ericoid mycorrhizal (EEM), orchid mycorrhizal (ORC), and non-mycorrhizal (NM). Panels **a** and **b** show these proportions for native (**a**) and naturalized (**b**) plant species; **c** and **d** show the difference between oceanic island and mainland in each type for native (**c**) and naturalized (**d**) species. Error bars represent standard errors of the means. All relevant statistics and sample sizes can be found in Supplementary Tables 2–6.

found a significant interaction between mycorrhizal type and distance, with the proportion of endemic AM species showing a faster rate of increase with distance compared to the other groups ($p < 0.001$, Fig. 4a and Supplementary Fig. 3). Specifically, the number of endemic species did not change with distance (Fig. 4b, $p = 0.74$), while the number of non-endemic AM species decreased strongly compared to other mycorrhizal types or NM plants (Fig. 4c, $p < 0.001$). Together, these results are consistent with the hypothesis that plants relying on AM fungi are more limited by the dispersal of their mutualists than plants associated with EEM fungi. This initial AM filter may allow for subsequent diversification of AM plant species in more distant islands. Finally, the proportion of AM to EEM plants varied with island

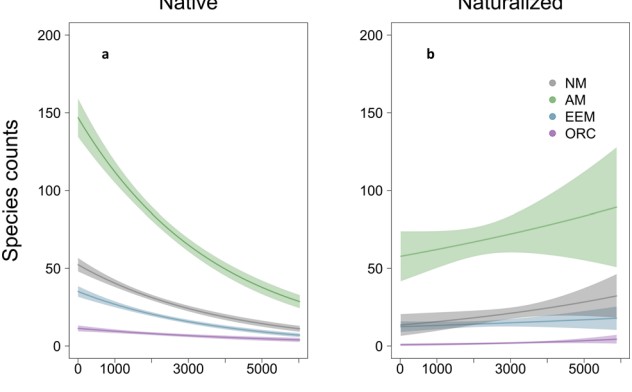

**Fig. 3 Distance patterns across mycorrhizal types in native and naturalized oceanic island floras.** The proportion of AM:NM plants in the native island flora decreases with oceanic island distance from the mainland (**a** estimate $= -0.034 \pm -0.006$, $p < 0.01$, $n = 325$; GLM), consistent with AM plants being differentially limited in colonization of far islands. In contrast, no patterns with distance are detectable in naturalized oceanic island floras (**b** estimate $= 0.034 \pm 0.005$, $p = 0.25$, $n = 105$; GLM). Confidence intervals represent 95% confidence intervals.

area (EEM:AM $p < 0.001$, Supplementary Table 3a M4; and EEM:NM $p < 0.001$, Supplementary Table 5a M4) and the proportion of endemism within these mycorrhizal types varied with area and elevation (Supplementary Figs. 4 and 5).

Native island floras showed a lower proportion of orchid plant species compared to mainlands. Specifically, compared to mainland floras, native island floras had approximately 3% lower orchid to non-mycorrhizal plant species (ORC:M $p = 0.06$, SI Table 3a M1; and ORC:NM $p < 0.001$, supplementary table 6a M1), consistent with the establishment limitation for orchids on islands. However, the proportion of orchid species increased with greater distance from the mainland as compared to both other mycorrhizal ($p < 0.001$, Supplementary Table 3a M4) as well as NM plant species ($p < 0.01$, Supplementary Table 6a M4), suggesting lower dispersal limitation.

**Environmental drivers of mycorrhizal species distributions in native floras.** For native mainland floras, variation in the proportion of mycorrhizal plant species was primarily explained by latitude and environmental variables. The proportion of EEM plant species increased non-linearly from the equator towards the poles (absolute latitude: EEM:AM $p < 0.001$, EEM:NM $p = 0.002$; absolute latitude squared: EM:AM $p < 0.001$, EEM:NM $p < 0.001$,

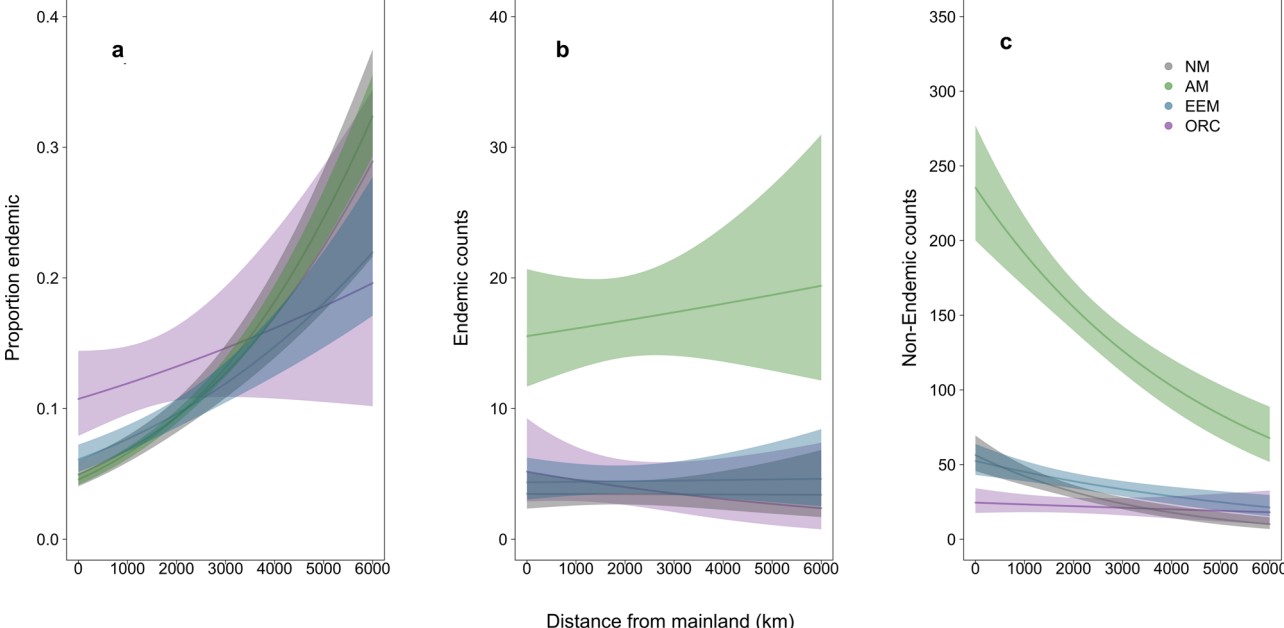

**Fig. 4 Variation in mycorrhizal types in oceanic island floras with distance from mainland source regions.** The proportion of plant species endemic to non-endemic increases most rapidly with distance for AM plant species (**a** estimate = 0.432 ± 0.063, $p < 0.001$, $n = 254$; GLM). The number of endemic AM species does not change with distance (**b** estimate 0.048 ± 0.147, $p = 0.74$, $n = 254$; GLM). The non-endemic species for AM decreases most strongly compared to other mycorrhizal types and to NM plants (**c** estimate = −0.265 ± 0.067, $p < 0.001$, $n = 254$; GLM). Confidence intervals represent 95% confidence intervals.

Supplementary Table 7). This may be indicative of the functional advantage of EEM symbioses in colder climates. AM and orchid plant species counts showed the strongest saturating declines with latitude compared to EEM and NM plant species (Fig. 5). This indicates that plants associating with AM or ORC fungi contribute more to the classical latitudinal plant species diversity gradient than EEM and NM plants. We note that the latitudinal gradient was present but diminished on islands (Supplementary Fig. 2).

**Environmental and anthropogenic drivers of mycorrhizal plant species distributions in naturalized floras.** Human-mediated plant naturalizations affected global plant biogeographical patterns influenced by mycorrhizal fungi (Figs. 1b and 2b, d). In the naturalized flora, we found evidence of an increase of approximately 1% of the representation of EEM relative to AM ($p < 0.001$, Supplementary Table 2a M2) and approximately 10% AM relative to NM plant species on islands ($p < 0.001$, Supplementary Table 4a M2). On oceanic islands, increasing urban land use was correlated with an increase in the proportion of AM plant species compared to NM plant species ($p = 0.02$), possibly due to horticultural introduction and early successional advantage of AM plant species. Further evidence of anthropogenic impacts on these biogeographical patterns was evident from the shift in drivers predicting the proportion of EEM plant species (EEM:AM, Supplementary Table 2a M5, M6) in naturalized floras. In naturalized island floras, we found evidence of humans overriding initial biogeographical patterns stemming from the mycorrhizal filter. Specifically, the effect of distance was reversed, with the proportion of AM plant species increasing with distance (Fig. 3b, AM:NM $p = 0.25$, Supplementary Table 4a M6). In mainland floras, increasing human land use was correlated with an increase in the proportion of EEM plant species (EEM:AM $p < 0.001$, EEM:NM $p < 0.001$), while population density was correlated with a decrease in the proportion of EEM plant species

(EEM:AM $p = 0.03$, Supplementary Table 2a M5; and EEM:NM $p < 0.001$, Supplementary Table 5a M5).

## Discussion
We found evidence that oceanic island angiosperm floras have different proportions of mycorrhizal plant species relative to mainlands. This 'disharmony' is consistent with the biology of mycorrhizal symbionts influencing the strength of a mycorrhizal filter during plant colonization of oceanic islands[9,10]. We found that native floras of oceanic islands worldwide had a lower proportion of plants that associate with AM fungi compared to mainland floras. Moreover, the proportion of AM plant species decreased with island isolation and this effect was particularly strong for non-endemic species. Together, these results are consistent with access to AM fungi limiting plant establishment on oceanic islands, as expected from their lower dispersal ability and inability to grow independently of their host. Limited AM plant colonization has led to an emergent pattern of insular disharmony in mycorrhizal species' types. However, the proportion of endemism of AM plants increases with island isolation, consistent with an evolutionary response of AM plants to this island disharmony.

We found consistent evidence of the legacy of dispersal limitation of plant species associating with AM fungi in contemporary native insular floras. These effects were evident in oceanic island floras, but not in floras of non-oceanic islands that were once connected to mainlands (Supplementary Table 2 M1a), supporting that the difference is in colonization rather than in rates of extinction or speciation. The persistence of these differences in contemporary native floras is remarkable, given the clear evidence that AM fungi do eventually colonize islands[32] and the thousands to millions of years required for secondary colonization and/or diversification to reverse initial differences. In fact, our results suggest that AM plant species have disproportionately high diversification rates. These AM plant species display higher endemism on distant islands, possibly due to the expanded

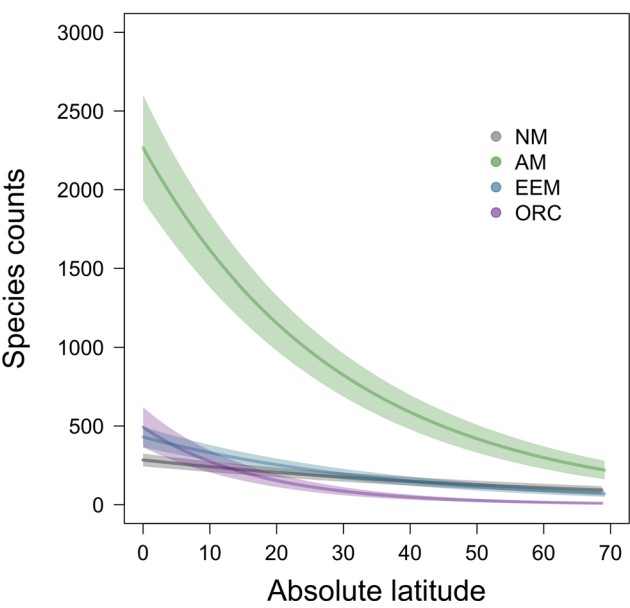

**Fig. 5 Latitudinal patterns across mycorrhizal types in native mainland plant species.** The latitudinal plant species gradient is strongly influenced by arbuscular (AM) and orchid (ORC) mycorrhizal plant species. In mainland regions, the proportion of mycorrhizal plant species decreases with absolute latitude (degree from equator) more strongly for arbuscular mycorrhizal (green line: absolute latitude estimate = −0.042 ± 0.080, $p$ = 0.60, $n$ = 515; squared latitude estimate −0.414 ± 0.080, $p$ < 0.001, $n$ = 515; GLM), than for ectomycorrhizal and ericoid (EEM) plant species (blue line: absolute latitude estimate = 0.051 ± 0.082, $p$ = 0.54, $n$ = 515; squared latitude estimate = −0.400 ± 0.082, $p$ < 0.001, $n$ = 515; GLM) and orchid mycorrhizal plant species (purple line: absolute latitude estimate = −1.216 ± 0.144, $p$ < 0.001, $n$ = 486; squared latitude estimate = 0.283 ± 0.144, $p$ = 0.05, $n$ = 486). Non-mycorrhizal species counts plotted for reference (gray line: absolute latitude estimate = 0.132 ± 0.132, $p$ = 0.1, $n$ = 486; squared latitude estimate = −0.330 ± 0.080, $p$ < 0.001, $n$ = 486; GLM). Confidence intervals represent 95% confidence intervals.

evolutionary opportunities generated by limited AM plant colonization. Our analyses support the hypothesis that dispersal limitation of AM fungi on distant islands is a stronger limiting factor in plant colonization of islands than the higher specificity of EEM associations. While there is empirical evidence of dispersal limitation of symbionts being important to both EEM and AM plant species[22,33–35], our work suggests that AM plants are more susceptible to symbiont dispersal limitation. Limited AM fungal dispersal to islands is supported by analyses of AM fungal composition showing differential AM fungal species abundances on islands compared to mainland regions[32].

We found that plants associating with AM fungi and ORC fungi contributed more to the latitudinal plant species diversity gradient than EEM and NM plants. This result mirrors the well-established pattern of EM trees being relatively more abundant in boreal forests, which has been associated with functional advantages such as short-circuiting the decomposition pathway through direct organic N uptake of EM symbiosis in colder climates[28,36]. We suggest that this same functional difference may have contributed to the differential pattern in plant species richness of EM versus AM plants across latitude. The patterns seen in orchid species mirror the close association orchids have with the tropics, given that 69% of orchids are epiphytic, which highly limits their distribution outside of the tropics[37,38].

Orchid species are generally under-represented on islands as compared to mainland regions, consistent with a previous study on global patterns in orchid richness[39] and consistent with limitation through high specificity of this symbiosis. However, we found evidence for higher proportions of orchid species on distant islands, consistent with high dispersal ability of orchids. These contrasting results may reflect the high ORC fungal specificity limiting island colonization overall, while the potential high dispersal ability of ORC fungi may contribute to orchid establishment on distant islands. Alternatively, these patterns may be influenced by other aspects of the biology of the Orchidaceae. For example, orchids produce abundant, but very small dust-like seeds, and feature a high dependency on and specialization of pollinators, which could influence the colonization of islands[40]. In contrast to orchid species, AM and EEM plant species occur across the plant phylogeny, increasing confidence that biogeographic patterns can be attributed to mycorrhizal fungal traits.

Naturalized and native floras showed distinct patterns. On oceanic islands, the proportion of AM plant species in naturalized floras was higher than in native floras and increased with both isolation and land-use intensity. This may result from the co-introduction of AM plants and their symbionts through the movement of agricultural and horticultural plants with soil[32]. This co-introduction may overcome the barriers to the establishment of AM plants on islands and allow them to fill in niche space left unfilled by the mycorrhizal filter[10]. On mainlands, however, higher land-use intensity has led to a greater proportion of EEM plant species, possibly due to non-native species in street planting and plantation use of EEM trees, which subsequently naturalized.

Our data support a legacy of arbuscular mycorrhizal (AM) fungi acting as a stronger filter on the initial colonization of islands compared to ectomycorrhizal and ericoid (EEM) and orchid (ORC) fungi, as AM plant species are under-represented in native island floras, and this effect increases with distance from the mainland, particularly for non-endemic plant species. These patterns are consistent with expectations of limited potential for colonization of newly formed islands by AM fungi due to limited dispersal ability and obligate host-dependence. We also find evidence of higher diversification rates of AM plants in response to the disharmony generated by the AM colonization filter. In native mainland floras, AM and orchid plant species contribute more strongly to the latitudinal plant species diversity gradient than EEM and NM plant species. Finally, anthropogenic impacts are diluting and, in some cases, reversing these biogeographical patterns. This work provides strong evidence that the major types of mycorrhizal fungi differentially influence plant colonization of islands, with subsequent effects on plant diversification and invasion risks.

## Methods

**Plant distribution data, mycorrhizal status, and explanatory variables.** Plant species occurrence data (for mostly administratively defined regions such as countries and provinces or islands), native status (native versus naturalized), and explanatory variables with regional characteristics were extracted from the Global Naturalized Alien Flora, GloNAF[41,42], and from the Global Inventory of Floras and Traits v 1.0, GIFT[43], databases. From the GloNAF database, we included only well-documented regions for which it was estimated that at least 50% of the naturalized species occurring in the given region were recorded. From the GIFT database, we used all regions for which checklists of native angiosperms were available. When there were overlapping regions, the smaller regions were kept if >100 km² for mainland regions; for islands, the smaller units were always preferred. Finally, we removed islands for which island geology (i.e. volcanic, floor, shelf, fragment, etc.) was undetermined.

The mycorrhizal status of plant species included in this study was determined by assigning each species to its plant family according to The Plant List[44], incorporating the classification from APG IV[45]. Following methods from Delavaux et al.[10], we relied on published family proportions of plant species' mycorrhizal status to assign mycorrhizal status proportions to the regional plant assemblages.

We used three review papers to determine a consensus proportion of mycorrhizal status per plant family[46–48]. Different classifications and proportions between the reference papers were accounted for by using the average proportion for each mycorrhizal type across the three references. While concerns have been raised over incorrect classification in these reviews[49,50], they cannot be addressed at this time due to the lack of species-specific corrections. Errors are mostly associated with EEM plant species, while most of our database is composed of AM plant species. We classified plant mycorrhizal status into four major types: arbuscular mycorrhizal (AM), ecto- and ericoid mycorrhizal (EEM), orchid mycorrhizal (ORC; occurs in Orchidaceae), and non-mycorrhizal (NM). We do not have a priori reason to expect differential colonization ability of ectomycorrhizal or ericoid mycorrhizal fungi because the distinction between these two groups is no longer clear. Specifically, molecular work has placed the model ericoid species *Hymenoscyphus ericae* into a larger fungal complex containing EM fungi, with the suggestion that these two groups of mycorrhizal fungi may constitute a single guild[51,52]. Therefore, we combine these mycorrhizal groups into one functional group in our analyses (EEM). We incorporated ambiguous classifications of mycorrhizal status (AMEM and AMNM), representing species found with both specified statuses, by running separate analyses, assigning species to either potential type. The full table of families and corresponding consensus proportions of mycorrhizal status can be found in supplementary data 1.

Explanatory variables for each region were extracted from the GIFT database. For details of environmental data collection, see Weigelt et al.[43]. Explanatory variables included land type (mainland or oceanic island), absolute latitude and longitude of the region's centroid, area (km²), mean annual temperature (°C) and mean annual precipitation[53] (mm), elevational range[54] (difference between lowest and highest elevation in m), human population density[55] (n/km²) and human land use. Human land use was calculated by combining two land-use metrics, cultivated and managed vegetation and urban land use area, as a sum followed by natural log transformation[56] (km²). When elevation range was unknown or reported as zero from aerial elevation maps, we assigned an elevation of 1 m as a minimum necessary elevation. For islands, we also included island distance to the nearest mainland (km) as a measure of geographical isolation[57] and island geological age. It is important to note that age reflects geological age, and not biological age, and so may not accurately reflect time from start of plant colonization. Data for endemism analyses represent a subset of islands for which endemism data were known.

We considered non-oceanic islands as oceanic islands if they were covered with ice (at least 80%) during the last glacial maximum[56], because they resemble newly formed oceanic islands after the plant and fungal communities were exterminated by glaciation. Before running models, we removed regions where there was a zero in total calculated species counts within any mycorrhizal type in a particular region. We removed these regions because these zero values may result from limited knowledge of mycorrhizal status of locally abundant plant families. We did not remove regions with a zero total for ORC as the orchid mycorrhizal association occurs only in Orchidaceae, which are likely to be correctly enumerated; therefore, we can reasonably assume that false zeros were unlikely for this mutualism.

**Statistics and reproducibility**. To investigate patterns of mycorrhizal plant distributions, we first used a multinomial logistic regression analogous to those described below to test how land type predicted mycorrhizal type of plant species; we found that the proportion of AM relative to NM plant species was reduced, while proportion of EEM relative to NM increased on oceanic islands compared to mainlands ($p < 0.001$, SI Table 1). While this analysis has the advantage of incorporating all mycorrhizal types simultaneously, it cannot account for non-independence of nearby islands (i.e., cannot include random effects). In order to account appropriately for the non-independence of geographically proximal islands, we used mixed models that correct for non-independence due to spatial proximity for a series of orthogonal comparisons (relative species number of pairs of mycorrhizal groupings) corresponding to the following comparisons: ectomycorrhizal to arbuscular mycorrhizal plant species (EEM:AM) and orchid mycorrhizal to all other types of mycorrhizal plant species (M), including arbuscular, ectomycorrhizal and ambiguous (AMEEM) mycorrhizal plant species (ORC:M). Next, to understand how these mycorrhizal types compare to non-mycorrhizal plant species (NM), we used additional comparisons of each mycorrhizal type compared to NM (EM:NM, AM:NM, and ORC:NM).

In our first set of models, we compared the species-richness patterns of plants with differing mycorrhizal associations. We ran models comparing: (i) EEM to AM plant species richness, (ii) ORC to M plant species richness, and (iii) each of the three mycorrhizal types (AM, EEM, ORC) compared to NM species richness. For each comparison, separate models were run for native and naturalized plants to predict plant species richness. In these generalized linear mixed effects models (GLMMs), we used a Poisson distribution because the response variable, species richness, was count data. The fixed effects were mycorrhizal status, land type (mainland non-oceanic island and oceanic islands) and their interaction; we also included the covariates of absolute latitude, the natural logarithm of area, the natural logarithm of elevation, and the natural logarithm of plant species richness. The random effects were region, nested within land type, and the interaction of region nested within land type with mycorrhizal status. These random terms control for the non-independence of individual plant species

records within floras, thereby providing general tests for differences in the proportion of mycorrhizal species across the floras of the different land types. The sample size ($n$) in these models represents a unique regional combination of native status (native or naturalized) and mycorrhizal status (reported in corresponding model tables). To create Fig. 1, we converted these count estimates to proportions.

Next, to investigate geographical and environmental drivers of mycorrhizal status for native and naturalized plants in mainland and oceanic island floras, we ran models comparing the proportion of: (i) EEM to AM plant species, (ii) ORC to M plant species, and (iii) each of the three mycorrhizal types (AM, EEM, ORC) compared to NM plant species. We used a composite response variable with species richness of each of the two mycorrhizal groupings of interest to account for differences in species richness. For these analyses, we used generalized linear models (GLMs) with a logit link function, assuming a binomial distribution of the response variable. For the native mainland models, we included the natural logarithm of area, the natural logarithm of elevation range, mean annual precipitation, mean annual temperature, absolute latitude and squared latitude. For the native island models, we included the same six variables with the addition of island distance to the mainland. We initially explored models with island age, however, as (1) this effect was not significant, (2) inclusion of this predictor substantially reduced the number of regions in the model, and (3) it did not meaningfully change distance effects, we include results of models without island age in the manuscript. The choice of variables was informed by prior studies of their effects on this dataset[10,58] as well as other island biogeographic studies[59,60]. As the presence of naturalized species is likely to be driven by human activities, the naturalized mainland models and the naturalized island models included human population density in addition to the explanatory variables included in the corresponding models for native species. Results of these models are presented in Supplementary Tables 2 through 6. Sample size ($n$) in all models excluding M1 and M2 are true $n$ representing unique regions. Next, we ran models testing for the proportion of endemic species in native oceanic island floras, using a composite response variable with counts of endemic species and non-endemic species. For these analyses, we used generalized linear models (GLMs) with a logit link function, assuming a binomial distribution of the response variable. As covariates, we included the natural logarithm of area, the natural logarithm of elevation range, the natural logarithm of age, absolute latitude and squared latitude. All predictor variables in models were mean-scaled prior to analysis.

To explore linear and non-linear latitudinal relationships, we reran all models comparing the five proportions described above including only absolute latitude and absolute latitude squared as the independent predictor variables, mean-scaled prior to analysis. We also ran models to investigate anthropogenic drivers of mycorrhizal status in naturalized plants only. For these models, we included a combined variable of urban land-use area and cultivated and managed vegetation, 'human land use' (sum of both variables). To assess the robustness of our results given the uncertainty in mycorrhizal status assignment, we reran all models testing for all combinations of ambiguous mycorrhizal status. In the main manuscript, we report statistics from models specified in Tables designated *a* (e.g. Supplementary Table 2a) in Tables S2–S6.

Generally, overdispersion in GLMs was adequately corrected using a quasi-binomial or quasi-Poisson family model. However, for latitude models, a negative binomial GLM was necessary to correct for overdispersion. In addition, most of our model residuals showed spatial autocorrelation as tested using Moran's I, which is expected in global scale models with spatially clustered geographic regions. We corrected for this spatial autocorrelation by including a spatial autocovariate that incorporates a matrix of longitude and latitude coordinates of the regions[60] in the *spdep* package in R[61]. After checking for spatial autocorrelation in our corrected models, some models still showed some spatial autocorrelation (as determined through Moran's I), but all spatial autocorrelation was reduced substantially. Because the naturalized ORC models (ORC:M and ORC:NM) were highly zero-inflated, caution should be taken in interpretation of results. All analyses were done in R 3.4.1[62] in the *lme4* package[63].

## Data availability
Data for plant family mycorrhizal status are presented in supplementary data 1; data on regions used in the paper are presented in ref 9.

## Code availability
All statistical code is made available at the following link: https://github.com/c383d893/MycorrhizalTypes (https://doi.org/10.5281/zenodo.5179626)[64].

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

## Acknowledgements

We acknowledge funding support from the US NSF (DEB 1556664, OIA 1656006 and DBI 2027458 to J.D.B., P.S., and C.S.D.), National Geographical Society (WW-036ER-17 to C.S.D.), German DFG (MvK: project 264740629, HK: FOR2716 DynaCom), Czech Science Foundation (EXPRO grant no. 19-28807X to P.P. and J.P.), Czech Academy of Sciences (RVO 67985939 to J.P. and P.P.), and Austrian Science Foundation (I 3757-B29 to F.E.).

## Author contributions

C.S.D. and J.D.B. designed the study. C.S.D. and J.D.B. led the statistical analysis and writing, with continued substantial contributions from P.W., H.K, and R.J.W.; C.S.D. collected the mycorrhizal data. All other authors, including W.D., F.E., M.VK., C.K., J.P., P.P., A.S., M.W., A.T., P.S., and R.J.W, collected the remaining plant and environmental data and edited the manuscript.

## Competing interests

The authors declare no competing interests.
