## [Transparent Peer Review File · Communications Biology]

Reviewers' comments:

Reviewer #1 (Remarks to the Author):

[Note this is the text pasted in from the attached document, which has some bold headings and line numbers]

This manuscript looks at the proportion of plant species (both native and naturalized) that form different mycorrhizal associated in mainland vs islands. They find some evidence of an arbuscular mycorrhizal (AM) filter which reduces the proportion of AM species relative to other function groups. This effect is only present in oceanic islands among the native flora. Moreover, it seems to be countered by an increased abundance of AM species in the naturalized flora and an increased rate of endemism among AM species (which is also consistent with an AM filter).

Overall the manuscript leverages an impressive dataset and provides some strong conclusions. I think these could be of broad interest to ecologists. However, I do have some reservations about the presentation, analyses, and interpretation of data in the manuscript. In summary:

- 1) I found the writing long-winded and occasionally convoluted
- 2) the a priori hypotheses about the AM filter are not adequately justified from the literature
- 3) discussion of effect sizes are missing
- 4) the figures are difficult to read and are missing raw data (also missing the case-study figures, which run contrary to the other conclusions and are buried in the supplement)
- 5) I am not entirely clear that the statistical tests controlled for expected unequal variances caused by the smaller species diversity of islands

These concerns are amenable to revision. In some cases (particularly 5), it may be that I simply did not understand how the statistical tests are, indeed, robust to these effects. However it would be nice to see, perhaps with simulated data drawn from a binomial distribution, what the expected island vs mainland distributions (and differences between these distributions) might be under the null hypothesis. This would put the 5% reduction in the proportion of AM species into context.

I will say that the interpretations and main results sound imminently plausible to me. It is just that in the manuscripts current form I am not sure I believe in them.

Introduction

The first sentence (77) was toil and trouble (get it? It's a reference to the three "which's" and also to starting a manuscript with a sentence that explains three separate mechanisms behind island biogeography. It's not unreadable, but I'd consider breaking it into three).

The whole manuscript has long sentences with lots of commas. Often two long sentences are written to make the same point. The info is tangled inside. Here's an example (assuming you introduce us to your mycorrhizal types first and then move on to your hypotheses):

"We can construct two a priori sets of expectations for the relative strength of the mycorrhizal filter based on differences in the biology of different groups of mycorrhizal fungi. Arbuscular mycorrhizal (AM) fungi, form the most common type of mycorrhizae, are likely to be most limited in their ability to colonize islands prior to plants due to two life-history traits."

Arguably it isn't a reviewer's job to critique style, but I think the manuscript would benefit from extensive line-edits. A more concise version would also be more impactful.

A contraction:

99: AM fungi lack a means of aerial dispersal.

100: the viability of aerial spores is unknown.

101: Moreover, AM fungi are the only group that cannot grow without a host.

You mean that they cannot be cultured, correct? Because it is ambiguous here if you are implying that other mycorrhiza grow non-symbiotically throughout parts of their natural ranges.

102: Now you're explicit -- albeit in a parenthesis with an *exempli gratia*. You mean EM and ORC can be saprotrophs. But you have only one reference -13.

Except 13 links to the seminal Smith & Read book "Mycorrhizal Symbioses." However, 13 is supposed to establish the saprotrophic lifestyles of both ectomycorrhizal and orchid mycorrhizal fungi. These claims are too specific to distribute on such a large, general text. We need proof for EMF and proof for ORC separately from the literature. I suspect this will be a bid harder to come by.

I am familiar with

Smith & Finlay (a different Smith here). 2017. Growing evidence of facultative biotrophy in saprotrophic fungi. *New Phytologist* 215: 747-755.

They provide some evidence that wood decaying fungi can form symbioses with EM tree hosts -- but this does not extend to all fungal taxa that associate with EM trees. Functionally there are big differences between the enzymatic abilities of saprotrophs and EM fungi. I know less about ORCs.

This is an important part because you have only two functional reasons to suppose that AM should have different dispersal abilities than EM and ORCs -- AMs lack aerial spores and need a host. But you're not off to a convincing start: you contradict yourself about the lack of AM aerial spores and you fail to support that EM and ORCs can be free-living.

114: AM fungi have lower specificity than EM and ORC. Again -- reference 13! Smith & Read's big book. These are specific and eyebrow raising claims. They need to be sourced to the primary literature. S&R is usually cited for the most basic, general statements about the ubiquity of mycorrhizal fungi.

Regarding Latitudinal trends from the intro:

I am a little confused about how this background informs your study. Could you circle this back to your hypothesis about naturalization on islands. As in, you expect latitudinal differences in the role of the mycorrhizal filter.

I am also concerned about the references in this section. You state that EM fungi are more common in colder biomes that slow down decomposition, but then cite Gomes et al. (2019) .Global distribution patterns of mycoheterotrophy. It seems that the Steidinger et al. (2019) paper (the next ref, 20) is more explicitly about decomposition rates, but it is not cited as such.

Methods

322

Combining the ericoids and the EM may be defensible in the context of your analysis (and they both are "ecto-" in the sense that arbuscular are "endo-"). But it is also different from how others have done this. Perhaps consider a different two-letter designation for the EM + Ericoids that reflects this difference, lest this methodological detail is forgotten when people cite and discuss your study's main results and figures.

And throughout the manuscript...

You need to clearly distinguish the proportion of species and the proportion of individuals. Throughout this manuscript you are looking at the proportion of species. You are also excluding gymnosperms, the majority of which are EM. If you looked at the proportion of AM tree individuals in Figure 1 you would generate a strong latitudinal trend (EM in the boreal forests and AM in the tropics) – so it's important to note that the two things cannot be talked about interchangeably.

Results

152

You begin your results with a case study that fails to support your main conclusions. (Moreover, I had no priming for these case-studies, so I was confused – perhaps mention the case-studies in your intro if the methods is to be buried beneath the rest of the text). You provide a lot of reasons why these islands are not a good test – but this begs the questions of whether it belongs in the study at all.

Be honest, would you have cast shade over Rakata and Surtsey if they gelled with the rest of your story about AM vs EM and ORC? If AM fungi are not wind-dispersed at all, is 40 km from the mainland really trivial?

Decide whether these data are worthy or not; if they are worthy, please present them (at least initially) shorn of interpretation. Preferably in a figure.

163

I see lots of p-values and "significant" trends – but almost no mention of effect sizes or % variability explained. Invite them in, please. I want to hear what they have to say. It looks to me like AM plants comprise about 5% less of the native plant species diversity on islands relative to the mainland (with NM plants picking up the slack).

166

"Compared to mainland, native islands had sig lower AM than EM (p=really small)." But is this really the measurement we want to know? This is a contrast of contrasts (correct?). Meaning we are looking at whether the decrease in the proportion of AM species is different from the increase in EM species (or ORC or NM).

Forgive me if I miss something important about the stats, but don't we first want to compare a null model – where the expected difference within a group is zero?

A bigger question

In the construction of a null hypothesis, do we need to consider the sampling effect of island size? Meaning that there should be some difference between a contrast between two mainland regions vs a contrast between a mainland and an island? Wouldn't we expect more variance in the distribution of the means of groups with a smaller sample size (n, given, for example, by the species richness of the islands), just as we expect a wider distribution of mean values when we look at distributions taken over means with lower vs larger sample sizes?

To be clear – I'm not certain the authors did anything wrong. But I need some assurance that the analysis is robust to this sampling size / unequal variance effect. Is that 5% decline in the proportion of AM species large compared to the expected variability in the means that comes from averaging over small islands? And could we maybe incorporate this into the figures, rather than standard error bars?

Discussion

First paragraph – these sound like cool findings and consistent with your hypotheses.

244

Here we having the higher specificity thing with EM (see 13). Let's clear up whether higher EM specificity is a real thing (see comment above).

254

"short-circuiting" decomposition pathway. A little unclear. It sounds like you are talking about the Gadgil effect here but it is ambiguous and not necessary to make your point.

Figures

A few thoughts. Lots of these figures show smooth spline fits, overlaid on one another, without the raw points. This makes the model fit look very deterministic and wonderful but it hides the proportion of variability explained. I think we really need to see the data-messiness here to evaluate these figures. I am also wondering if you should log scale the y-axes for most of your species counts, given that your naturalization / filter stuff looks at proportional representation of species. This could linearize some of the relationships so we could compare the slopes.

Reviewer #2 (Remarks to the Author):

Interesting database mining study with some new findings, focusing on whether different mycorrhizal types correlate with the composition of island floras. Corrections and a few suggestions for improvement are outlined below.

1 ...biogeography of flowering plants (this is important to clarify upfront as dominant plants in boreo-temperate biomes are excluded, and there is evidence of non-vascular plants harboring significant mycorrhizal diversity globally, e.g. Proc B 285: 20181600).

63. ...ectomycorrhizal and ericoid (EM) and... (need to be explicit about this at least once in the abstract and once in early in the introduction, otherwise it is hidden until almost the end).

64. ...worldwide, including XX islands.

86. Provide reference(s) for "limit the colonization of mycorrhizal plant species".

95. Paragraph needs to mention horizontal transmission characterizes mycorrhizae.

102-104. This is no different between AMF and EM; most, if not all, ectomycorrhizal fungi are obligate biotrophs (Mycorrhiza DOI 10.1007/s00572-009-0274-x), i.e. they lack cellulases and in nature spores do not grow much at all (i.e. beyond a germination tube) without a host, like arbuscular mycorrhizal fungi. However, lifecycles are starkly different (asexual in AMF, requiring mating of two compatible spores for root colonization in basidiomycete EMF), numbers of propagules per individual and spore longevity/dormancy/viability may well differ too.

104. This statement about aerial dispersal merits references for ORC, if they exist.

178, 180. Can you mention how they "varied"?

260-264. Avoid repetitive unnecessary jargon, e.g. "ORC plant species" are "orchids".

292-3. Avoid overstatement, this study is about colonization of islands, not the rest.

303-4. Estimated how?

323. This statement is incorrect and reference 43 is misinterpreted.

395. This part about island age merits some discussion in the manuscript; why would age not play a role?

Camille S. Delavaux
PhD
Ecology and Evolutionary Biology
9 Takeru Higuchi Hall
2101 Constant Ave.
Lawrence KS, 66047
Email: camille.delavaux@ku.edu
Phone: (908) 892-9717 (cell)

June 29, 2021

We thank both reviewers for their constructive comments and suggestions that these have improved our manuscript. Please find our responses to each comment below.

*Sincerely and on behalf of our co-authors,
Camille S. Delavaux & James D. Bever*

Reviewer #1 (Remarks to the Author):

[Note this is the text pasted in from the attached document, which has some bold headings and line numbers]

This manuscript looks at the proportion of plant species (both native and naturalized) that form different mycorrhizal associated in mainland vs islands. They find some evidence of an arbuscular mycorrhizal (AM) filter which reduces the proportion of AM species relative to other function groups. This effect is only present in oceanic islands among the native flora. Moreover, it seems to be countered by an increased abundance of AM species in the naturalized flora and an increased rate of endemism among AM species (which is also consistent with an AM filter).

Overall the manuscript leverages an impressive dataset and provides some strong conclusions. I think these could be of broad interest to ecologists. However, I do have some reservations about the presentation, analyses, and interpretation of data in the manuscript. In summary:

- 1) I found the writing long-winded and occasionally convoluted
- 2) the a priori hypotheses about the AM filter are not adequately justified from the literature
- 3) discussion of effect sizes are missing
- 4) the figures are difficult to read and are missing raw data (also missing the case-study figures, which run contrary to the other conclusions and are buried in the supplement)
- 5) I am not entirely clear that the statistical tests controlled for expected unequal variances caused by the smaller species diversity of islands

These concerns are amenable to revision. In some cases (particularly 5), it may be that I simply did not understand how the statistical tests are, indeed, robust to these effects. However it would be nice to see, perhaps with simulated data drawn from a binomial distribution, what the expected island vs mainland distributions (and differences between these distributions) might be under the null hypothesis. This would put the 5% reduction in the proportion of AM species into context.

I will say that the interpretations and main results sound imminently plausible to me. It is just that in the manuscripts current form I am not sure I believe in them.

Thank you for your comments and suggestions. We have condensed and clarified the writing, supplemented references to lay a stronger foundation to our proposed hypotheses, included the effect sizes of the major results

in the text, and described how our statistical approach accounts for unexpected variances caused by differences in species diversity on islands compared to mainlands. We further followed your suggestion and simulated data from the binomial distribution to determine the expected island and mainland distributions. These confirm that the results we obtained are robust to differences in sample sizes across our observations. We have addressed these concerns in detail following each comment below.

Introduction

The first sentence (77) was toil and trouble (get it? It's a reference to the three "which's" and also to starting a manuscript with a sentence that explains three separate mechanisms behind island biogeography. It's not unreadable, but I'd consider breaking it into three).

Thank you. We have edited this sentence to be more straightforward and concise (L 83-88). It now reads as follows:

Classical island biogeography recognizes that species richness results from the net effects of immigration, extinction and speciation. These biogeographical rates have been primarily linked to abiotic features of islands: immigration decreases with isolation, extinction decreases with island size¹, and speciation increases with island size and isolation²⁻³. However, only a limited number of case studies have addressed how biotic interactions influence colonization, extinction, and speciation probabilities on islands⁴⁻⁶, and generalizations are difficult.

The whole manuscript has long sentences with lots of commas. Often two long sentences are written to make the same point. The info is tangled inside. Here's an example (assuming you introduce us to your mycorrhizal types first and then move on to your hypotheses):

"We can construct two a priori sets of expectations for the relative strength of the mycorrhizal filter based on differences in the biology of different groups of mycorrhizal fungi. Arbuscular mycorrhizal (AM) fungi, form the most common type of mycorrhizae, are likely to be most limited in their ability to colonize islands prior to plants due to two life-history traits."

Arguably it isn't a reviewer's job to critique style, but I think the manuscript would benefit from extensive line-edits. A more concise version would also be more impactful.

We agree that we should introduce the mycorrhizal types first and then move on to our expectations. We have added a sentence clarifying the three groups of mycorrhizal fungi we are investigating in our manuscript, enabling us to simplify the referenced sentence (L 99-102). This now reads as follows:

Here, we test for differences in the strength of the mycorrhizal filter across plant species associating with the three major mycorrhizal types: arbuscular mycorrhizal (AM) fungi, the most common type of mycorrhizae, ectomycorrhizal and ericoid mycorrhizal (EEM) fungi, and orchid mycorrhizal (ORC) fungi.

We further agree that certain sentences could be simplified for greater clarity and impact. We have gone through the manuscript to simplify and separate longer sentences into several sentences.

Here is an example:

Original: In fact, our results suggest that plants that associate with AM fungi having disproportionately high diversification rates, as they display high endemism on distant islands, consistent with expanded opportunity due to the disharmony generated by limited colonization.

New (L 248-251): In fact, our results suggest that AM plant species have disproportionately high diversification rates. These AM plant species display higher endemism on distant islands, possibly due to the expanded evolutionary opportunities generated by limited AM plant colonization.

A contraction:

99: AM fungi lack a means of aerial dispersal.

We have changed our text to clarify that AM fungi do not have adaptations for aerial dispersal like EEM and ORC fungi can (L 105-113). This now reads as follows:

AM fungi are likely to be most limited in their ability to colonize islands prior to host plants due to two life-history traits. First, AM fungi lack adaptations for aerial dispersal (but see Chaudhary et al. 2020¹¹). Moreover, AM fungi are the only group of mycorrhizal fungi that cannot grow independently of their hosts¹². In contrast, other types of mycorrhizal fungi, including EEM, as well as ORC fungi, can have adaptations for aerial dispersal of spores^{13,14} and can grow independently of their host through saprophytic activity^{12,15-20}. We therefore expect EEM and ORC fungi to be better able to establish on islands prior to their hosts compared to AM fungi, and EEM and orchid plant species to be less impacted by the mycorrhizal filter than AM plant species²¹.

100: the viability of aerial spores is unknown.

We have removed this clause as it was not necessary here. The major point is that although there may be some aerial dispersal of AM fungi, they do not have an adaptation to do so.

101: Moreover, AM fungi are the only group that cannot grow without a host.

You mean that they cannot be cultured, correct? Because it is ambiguous here if you are implying that other mycorrhiza grow non-symbiotically throughout parts of their natural ranges.

This is correct, AM fungi are the only group of mycorrhizal fungi that require a plant host to grow. They therefore require a host to be cultured in a greenhouse setting but also in the field. Other types of mycorrhizal fungi are able to grow saprotrophically in the absence of a host. We agree that additional references are important to confirm the saprophytic growth capabilities of EEM (ectomycorrhizal and ericoid) and ORC fungi and have added relevant references here to support this (Shah et al. 2016, Read & Perez-Moreno 2003, Martino et al. 2018, Selosse et al. 2010, McCormik et al. 2012).

This section now reads as follows (L 108-110):

In contrast, other types of mycorrhizal fungi, including EEM, as well as ORC fungi, can have adaptations for aerial dispersal of spores^{13,14} and can grow independently of their host through saprophytic activity^{12,15-20}.

102: Now you're explicit -- albeit in a parenthesis with an exempli gratia. You mean EM and ORC can be saprotrophs. But you have only one reference -13.

[smith and read specific]

Except 13 links to the seminal Smith & Read book “Mycorrhizal Symbioses.” However, 13 is supposed to establish the saprotrophic lifestyles of both ectomycorrhizal and orchid mycorrhizal fungi. These claims are too specific to distribute on such a large, general text. We need proof for EMF and proof for ORC separately from the literature. I suspect this will be a bid harder to come by.

I am familiar with

Smith & Finlay (a different Smith here). 2017. Growing evidence of facultative biotrophy in saprotrophic fungi. *New Phytologist* 215: 747-755.

They provide some evidence that wood decaying fungi can form symbioses with EM tree hosts -- but this does not extend to all fungal taxa that associate with EM trees. Functionally there are big differences between the enzymatic abilities of saprotrophs and EM fungi. I know less about ORCs.

This is an important part because you have only two functional reasons to suppose that AM should have different dispersal abilities than EM and ORCS – AMs lack aerial spores and need a host. But you’re not off to a convincing start: you contradict yourself about the lack of AM aerial spores and you fail to support that EM and ORCs can be free-living.

We agree that this section needed more support and a clearer justification of expected differences between AM and EEM/ORC fungi. We refer to the Chaudhary paper that shows that small spored AM fungi can be blown in the wind with soil erosion to be complete. This does not provide evidence that the spores are alive and definitely does not provide any reason to think that AM fungi have adaptations for aerial dispersal. This is in strong contrast to the work showing that some EM fungi do have adaptations for aerial dispersal. Similarly, it is common practice to grow EM fungi on nutrient media, reflecting their ability to grow on external sugars, while AM fungi can only be grown on live plant roots. Following your recommendation, we have revised this section, adding Smith and Finlay (2017) and providing additional references (Shah et al. 2016, Read & Perez-Moreno 2003, Martino et al. 2018, Selosse et al. 2010, McCormik et al. 2012). Please see expanded references supporting this statement in response to the previous comment above.

114: AM fungi have lower specificity than EM and ORC. Again – reference 13! Smith & Read’s big book. These are specific and eyebrow raising claims. They need to be sourced to the primary literature. S&R is usually cited for the most basic, general statements about the ubiquity of mycorrhizal fungi.

We agree that more references are needed and have added several references here to support the difference in specificity between the mycorrhizal types. Specifically, we have added a paper by Chaudhary et al. (2016) showing broad phylogenetic associations of AM fungi, but less so in EM fungi. We also added a paper by van der Heijden et al. (2015) showing that the ratio of fungi to plants is much lower in the AM symbiosis, leading to expected lower specificity in AM plant-fungal relationships. This paper also has a figure with (increasing) specificity across AM, EM and ORC fungi (L 114-127).

Regarding Latitudinal trends from the intro:

I am a little confused about how this background informs your study. Could you circle this back to your hypothesis about naturalization on islands. As in, you expect latitudinal differences in the role of the mycorrhizal filter.

Thank you for this comment. This section is in reference to our tests of what environmental drivers impact proportions of mycorrhizal types. We have added a hypothesis here (L 128-143). This now reads as follows:

Besides acting as a filter on colonization, the types of mycorrhizal associations may influence the global distribution patterns of plant species through functional differences, providing additional hypotheses

relevant to global biogeography. For instance, AM fungi are thought to be most effective at facilitating access to relatively immobile resources such as inorganic phosphorus and nitrogen released by saprotrophs, and EM fungi are commonly thought to be able to better access organic nitrogen²⁶, potentially short-circuiting the decomposition pathway. This function is assumed to be particularly important in colder climates where decomposition is slow. These differences underlie arguments for the dominance of EEM plant species in colder climates¹⁶. Recent analyses built on assumptions of the ecological differences in AM and EEM symbioses predict extant patterns of mycorrhizal types in forests, with greater dominance of AM plant species near the equator and greater dominance of EEM plant species closer to the poles^{27,28}. Therefore, we expect the biogeography of AM and EM mycorrhizal plant species to be driven in part by temperature and precipitation, important decomposition-related environmental variables that vary with latitudes. Predictions based on functional differences of ORC fungi are difficult, as the associated plants can be parasitic rather than mutualistic with their fungi^{29,30}.

I am also concerned about the references in this section. You state that EM fungi are more common in colder biomes that slow down decomposition, but then cite Gomes et al. (2019) .Global distribution patterns of mycoheterotrophy. It seems that the Steidinger et al. (2019) paper (the next ref, 20) is more explicitly about decomposition rates, but it is not cited as such.

We agree that more careful citation here is needed. We first cite a paper supporting functional differences between AM and EM, and then cite two papers showing global biogeographical patterns and their link to these functional differences. Specifically, we cite Read and Perez-Moreno (2013) and Phillips as a functional explanation behind this difference (L 135-136):

These differences underlie arguments for the dominance of EM plant species in colder climates¹⁶.

We then refer to both Steidinger et al. (2019) and Gomes et al. (2019), which show global biogeographical patterns in support of these functional differences. Steidinger et al. describe the global pattern of greater EM toward the poles and hypothesize that this distribution of mycorrhizal types across forests is related to climatic controls over decomposition. Gomes et al. look at the mycoheterotrophic subset of AM and EM plant species, but again show this broad pattern of a shift from AM dominated to EM dominated forests from low to high latitudes (L 136-139):

Recent analyses built on assumptions of the ecological differences in AM and EM symbioses predict extant patterns of mycorrhizal types in forests, with greater dominance of AM plant species near the equator and greater dominance of EM plant species closer to the poles^{26,27}.

Methods

322

Combining the ericoids and the EM may be defensible in the context of your analysis (and they both are “ecto-“ in the sense that arbuscular are “endo-“). But it is also different from how others have done this. Perhaps consider a different two-letter designation for the EM + Ericoids that reflects this difference, lest this methodological detail is forgotten when people cite and discuss your study’s main results and figures.

We agree that this important to clarify. We have therefore changed our EM designation which includes both ecto- and ericoid mycorrhizas to ‘EEM’ throughout the manuscript; we have also clarified this grouping in the Abstract and Introduction:

Abstract (L 67-69):

Plant colonization of islands may be limited by the availability of symbionts, particularly arbuscular mycorrhizal (AM) fungi, which have limited dispersal ability compared to ectomycorrhizal and ericoid (EEM) and orchid mycorrhizal (ORC) fungi.

Introduction (L 99-102):

Here, we test for differences in the strength of the mycorrhizal filter across plant species associating with the three major mycorrhizal types: arbuscular mycorrhizal (AM) fungi, the most common type of mycorrhizae, ectomycorrhizal and ericoid mycorrhizal (EEM) fungi and orchid mycorrhizal (ORC) fungi.

And throughout the manuscript...

You need to clearly distinguish the proportion of species and the proportion of individuals. Throughout this manuscript you are looking at the proportion of species. You are also excluding gymnosperms, the majority of which are EM. If you looked at the proportion of AM tree individuals in Figure 1 you would generate a strong latitudinal trend (EM in the boreal forests and AM in the tropics) – so it's important to note that the two things cannot be talked about interchangeably.

Thank you for noting this. We have noted at the end of the Introduction, the beginning of the Results and in the Discussion that these results are for angiosperm species only. Throughout the manuscript, we are using a dataset consisting of species checklists, so we are always examining the proportions of plant species. We agree that this should be made clear and have changed our uses of “AM plants” and “EEM plants” to “AM plant species” and “EEM plant species”.

Abstract (L 69-70):

We tested for such differential island colonization within contemporary angiosperm floras worldwide.

End of Introduction (L 144-145):

Here, we explore biogeographical patterns of angiosperm species that associate with different types of mycorrhizal fungi.

Results (L 168-169):

Across oceanic islands globally, we found support for dispersal limitation of native angiosperm plant species that associate with AM fungi (Fig. 1, Fig. 2, SI Tables 1-6).

Discussion (L 229-230):

We found evidence that oceanic island angiosperm floras have different proportions of mycorrhizal plant species relative to mainlands.

Results

152

You begin your results with a case study that fails to support your main conclusions. (Moreover, I had no priming for these case-studies, so I was confused – perhaps mention the case-studies in your intro if the methods is to be buried beneath the rest of the text). You provide a lot of reasons why these islands are not a good test – but this begs the questions of whether it belongs in the study at all.

Be honest, would you have cast shade over Rakata and Surtsey if they gelled with the rest of your story about AM vs EM and ORC? If AM fungi are not wind-dispersed at all, is 40 km from the mainland really trivial?

Decide whether these data are worthy or not; if they are worthy, please present them (at least initially) shorn of interpretation. Preferably in a figure.

We agree that this analysis unnecessarily complicates our study and does not need to be included. We have therefore removed it from the manuscript.

163

I see lots of p-values and “significant” trends – but almost no mention of effect sizes or % variability explained. Invite them in, please. I want to hear what they have to say. It looks to me like AM plants comprise about 5% less of the native plant species diversity on islands relative to the mainland (with NM plants picking up the slack).

We agree with the reviewer on the importance of presenting the effect sizes. We have reported effect sizes from all analyses in the supplementary tables. However, as these effect sizes are in units of logits, they are not easily understood by many readers. We have therefore made all figures in units of species number. In Figure 2, we have also made C and D explicitly as the change in proportion to facilitate understanding of the magnitude of the effects that we are reporting. In response to the reviewer’s concerns, we now also report the magnitude of the effects in the results text; this now reads as follows:

(L 168-172)

Across oceanic islands globally, we found support for dispersal limitation of native angiosperm plant species that associate with AM fungi (Fig. 1, Fig. 2, SI Tables 1-6). Specifically, compared to mainland floras, we found that native island floras had approximately 5% lower proportion of AM than EEM (EEM:AM $p < 0.001$; SI Table 2a M1) and 10% lower AM than NM plant species (AM:NM $p < 0.001$; SI Table 4a M1).

(L 187-190)

Native island floras showed a lower proportion of orchid species compared to mainlands. Specifically, compared to mainland floras, native island floras had approximately 3% lower orchid to non-mycorrhizal plant species (ORC:M $p = 0.06$, SI Table 3a M1; and ORC:NM $p < 0.001$, SI Table 6a M1), consistent with the establishment limitation for orchids on islands.

(L 209-212)

Human-mediated plant naturalizations affected global plant biogeographical patterns influenced by mycorrhizal fungi (Fig. 2). In the naturalized flora, we found evidence of an increase of approximately 1% of the representation of EM relative to AM ($p < 0.001$, SI Table 2a M2) and approximately 10% AM relative to NM plant species on islands ($p < 0.001$, SI Table 4a M2).

166

“Compared to mainland, native islands had sig lower AM than EM (p =really small).” But is this really the measurement we want to know? This is a contrast of contrasts (correct?). Meaning we are looking at whether the decrease in the proportion of AM species is different from the increase in EM species (or ORC or NM).

Forgive me if I miss something important about the stats, but don’t we first want to compare a null model – where the expected difference within a group is zero?

We set out to test whether the different categories of mycorrhizal fungi differentially limit colonization of islands by plants. This necessarily translates into a statistical contrast of differences of differences as you correctly identify.

A bigger question

In the construction of a null hypothesis, do we need to consider the sampling effect of island size? Meaning that there should be some difference between a contrast between two mainland regions vs a contrast between a mainland and an island? Wouldn't we expect more variance in the distribution of the means of groups with a smaller sample size (n, given, for example, by the species richness of the islands), just as we expect a wider distribution of mean values when we look at distributions taken over means with lower vs larger sample sizes?

To be clear – I'm not certain the authors did anything wrong. But I need some assurance that the analysis is robust to this sampling size / unequal variance effect. Is that 5% decline in the proportion of AM species large compared to the expected variability in the means that comes from averaging over small islands? And could we maybe incorporate this into the figures, rather than standard error bars?

Thank you for noting this. Our models account for reduced variance with small means by identifying the Poisson error distribution (or in subsequent models identifying the response variable as a combined response variable). To illustrate that our model is robust to unequal variance, we followed your suggestion and generated data with the binomial distribution with the means of AM and EM species on islands and mainlands. We ran three simulations. In the first, we set the proportion of AM:EM equal to 0.5 and generated 1000 mainlands and 1000 islands, each with 100 species. Here, we find no interaction between entity_class (island or mainland) and mycorrhizal status (myc, AM or EEM). Next, we simulated data with the proportion AM:EM equal to 0.5, but the number of islands and mainlands and species therein set to the averages in our real dataset. Here, again we find no interaction between entity_class and myc; the significant effect of the mainland is expected due to the higher number of mainlands than islands. Importantly, this null model does not show the result we find in our analyses, showing that our analyses are robust to differences in sample size. Finally, when we set the proportion AM:EEM, the number of islands and mainlands and species therein to the averages in our dataset, we do find a significant interaction between entity_class and mycorrhizal status. Overall, this illustrates that we obtain our major finding (interaction between entity_class and myc) with the simulated dataset taking into account the real means, but not with two null models (either with an equal number of locations and species per location for mainland and island or with a number of locations and species equivalent to the means in our real data).

Please find the simulation results below:

*Model: lmer(myc1~myc*entity_class2+(1/entity_class2:entity_ID) +(1/entity_class2:entity_ID:myc),family = poisson, data = binomdat)*

SIM. 1: prop AM set to 0.5; 1000 mainlands and 1000 islands, each with 100 species

	Estimate	Std. Error	z value	Pr(> z)
(Intercept)	5.521121	0.002000	2760.091	<2e-16 ***
mycEEM	0.000680	0.002828	0.240	0.810
entity_class2mainland	-0.001482	0.002830	-0.524	0.601
mycEEM:entity_class2mainland	0.002960	0.004000	0.740	0.459

SIM. 2: prop AM set to 0.5; number of locations and species set to island or mainland means

	Estimate	Std. Error	z value	Pr(> z)
(Intercept)	4.7959548	0.0036566	1311.589	<2e-16 ***
mycEEM	-0.0003276	0.0051716	-0.063	0.949
entity_class2mainland	1.5316296	0.0039164	391.083	<2e-16 ***
mycEEM:entity_class2mainland	-0.0007541	0.0055392	-0.136	0.892

SIM. 3: prop AM, number of locations and species set to island or mainland means

	Estimate	Std. Error	z value	Pr(> z)
(Intercept)	3.798430	0.006021	630.84	<2e-16 ***
mycEEM	1.486645	0.006667	222.97	<2e-16 ***
entity_class2mainland	1.454904	0.006482	224.46	<2e-16 ***
mycEEM:entity_class2mainland	0.092835	0.007169	12.95	<2e-16 ***

Discussion

First paragraph – these sound like cool findings and consistent with your hypotheses.

Thank you.

244

Here we having the higher specificity thing with EM (see 13). Let’s clear up whether higher EM specificity is a real thing (see comment above).

Thank you. We have added relevant references to show that EM and ORC have higher specificity than AM here (Chaudhary et al. 2016; van der Heijden et al. 2015; L 114-127).

254

“short-circuiting” decomposition pathway. A little unclear. It sounds like you are talking about the Gadgil effect here but it is ambiguous and not necessary to make your point.

We agree that clarifying what we mean here is important and have therefore elaborated on what we are referring to. EM fungi can uptake nutrients directly from organic matter, while AM fungi uptake nutrients released from saprophytic fungi. This results in EM fungi bypassing the need for saprotrophs and decomposition. We have added two additional references here as well (Phillips et al. 2013, Read & Perez-Moreno 2003). This section now reads as follows (L 258-262):

We found that plants associating with AM fungi and ORC fungi contributed more to the latitudinal plant species diversity gradient than EEM and NM plants. This result mirrors the well-established pattern of EM trees being relatively more abundant in boreal forests, which has been associated with functional advantages such as short-circuiting the decomposition pathway through direct organic N uptake of EM symbiosis in colder climates^{26,34}.

Figures

A few thoughts. Lots of these figures show smooth spline fits, overlaid on one another, without the raw points. This makes the model fit look very deterministic and wonderful but it hides the proportion of variability explained. I think we really need to see the data-messiness here to evaluate these figures. I am also wondering if you should log scale the y-axes for most of your species counts, given that your naturalization / filter stuff looks at proportional representation of species. This could linearize some of the relationships so we could compare the slopes.

We agree that including the data with the figures is important for assessment by the reader. However, for our study, there are so much data that the data points on the edge of the distribution may be more evident to the human eye than the mass of points lost near the mean, thereby obscuring the general result. We believe the plots of the model fit with confidence intervals in our manuscript figures more clearly convey the study results and do not misrepresent results, and so we retain these in the main manuscript. We include figures with model residuals in the Supplementary Information.

Finally, the lines and confidence intervals plotted here are predicted results from the model (put x in, the model predicts y). For the variable of interest, we set all other variables in the model to their mean to calculate the response predicted for each change in the variable of interest. Although we understand the desire to linearize certain figures (e.g Fig. 5), taking the log of species would make some figures non-linear (e.g. Fig. 4). Further, because we are looking at the latitudinal diversity gradient in plant species as altered by mycorrhizal type, particularly in Fig. 5, we believe it is easier for the reader to have species counts on the y axis. To be consistent across figures, we keep species counts in the figures predicting species.

Reviewer #2 (Remarks to the Author):

Interesting database mining study with some new findings, focusing on whether different mycorrhizal types correlate with the composition of island floras. Corrections and a few suggestions for improvement are outlined below.

Thank you for your constructive comments and suggestions. They have helped us increase the clarity and impact of our manuscript. Please find our detailed responses below.

1 ...biogeography of flowering plants (this is important to clarify upfront as dominant plants in boreo-temperate biomes are excluded, and there is evidence of non-vascular plants harboring significant mycorrhizal diversity globally, e.g. Proc B 285: 20181600).

We agree and have specified that we are investigating these patterns in flowering plants/angiosperms only; we have added this to the abstract, the end of the Introduction, the start of the Results and the start of the Discussion.

Abstract (L 69-70):

We tested for such differential island colonization within contemporary angiosperm floras worldwide.

End of Introduction (L 144-145):

Here, we explore biogeographical patterns of angiosperm species that associate with different types of mycorrhizal fungi.

Results (L 168-169):

Across oceanic islands globally, we found support for dispersal limitation of native angiosperm plant species that associate with AM fungi (Fig. 1, Fig. 2, SI Tables 1-6).

Discussion (L 229-230):

We found evidence that oceanic island angiosperm floras have different proportions of mycorrhizal plant species relative to mainlands.

63. ...ectomycorrhizal and ericoid (EM) and... (need to be explicit about this at least once in the abstract and once in early in the introduction, otherwise it is hidden until almost the end).

We agree and have clarified this in the abstract and introduction. We have also changed our EM designation which includes both ecto- and ericoid mycorrhizas to 'EEM' throughout the manuscript.

Abstract (L 67-69):

Plant colonization of islands may be limited by the availability of symbionts, particularly arbuscular mycorrhizal (AM) fungi, which have limited dispersal ability compared to ectomycorrhizal and ericoid (EEM) and orchid mycorrhizal (ORC) fungi.

Introduction (L 99-102):

Here, we test for differences in the strength of the mycorrhizal filter across plant species associating with the three major mycorrhizal types: arbuscular mycorrhizal (AM) fungi, the most common type of

mycorrhizae, ectomycorrhizal and ericoid mycorrhizal (EEM) fungi and orchid mycorrhizal (ORC) fungi.

64. ...worldwide, including XX islands.

Number of islands included in each analysis is dependent on data available for each comparison between pairs of mycorrhizal categories. Therefore, we believe it would be misleading to put a specific number here. We do, however, note the sample size for each analysis in the supplementary information at the start of each table.

86. Provide reference(s) for "limit the colonization of mycorrhizal plant species".

We agree that a reference supporting the limitation of colonization of mycorrhizal plant species is important in this sentence. We then only cite the global analysis in the subsequent sentence, as it directly references this study. This section now reads as follows:

Many plant species are highly dependent on mycorrhizal fungi⁸, however, these fungi disperse independently of their plant hosts and therefore the absence of mycorrhizal fungi may limit the colonization of mycorrhizal plant species⁹. Indeed, a recent global analysis of native floras found both a lower proportion of mycorrhizal plant species on islands than on mainlands and a decrease in the proportion of mycorrhizal plant species in island floras with increasing isolation¹⁰, consistent with the operation of a mycorrhizal filter on plant colonization of islands.

95. Paragraph needs to mention horizontal transmission characterizes mycorrhizae.

We have clarified that fungi disperse independently of their host in the introduction. This now reads as follows (L 88-94):

Order of arrival, resulting in priority effects⁷, is likely to be particularly important for mutualisms. The mycorrhizal mutualisms formed between soil fungi and most plant species are prime candidates for priority effects. Many plant species are highly dependent on mycorrhizal fungi⁸, however these fungi disperse independently of their plant hosts and therefore the absence of mycorrhizal fungi may limit the colonization of mycorrhizal plant species⁹.

102-104. This is no different between AMF and EM; most, if not all, ectomycorrhizal fungi are obligate biotrophs (Mycorrhiza DOI 10.1007/s00572-009-0274-x), i.e. they lack cellulases and in nature spores do not grow much at all (i.e. beyond a germination tube) without a host, like arbuscular mycorrhizal fungi. However, lifecycles are starkly different (asexual in AMF, requiring mating of two compatible spores for root colonization in basidiomycete EMF), numbers of propagules per individual and spore longevity/dormancy/viability may well differ too.

We have clarified this to note that AM fungi lack adaptations for aerial dispersal, while EEM and ORC fungi can have adaptations for aerial dispersal. We acknowledge the important work showing that specialization on the mycorrhizal habit has come with some enzymatic losses that will decrease their efficiency as decomposers. However, EEM fungi can still grow on external sugars (as they are commonly maintained in the lab), which is still a sharp contrast to AM fungi, which cannot. We have additionally added several references supporting the saprotrophic activity of EEM and ORC fungi.

This now reads as follows(L 105-113):

AM fungi are likely to be most limited in their ability to colonize islands prior to host plants due to two life-history traits. First, AM fungi lack adaptations for aerial dispersal (but see Chaudhary et al. 2020¹¹). Moreover, AM fungi are the only group of mycorrhizal fungi that cannot grow independently of

their hosts¹². In contrast, other types of mycorrhizal fungi, including EEM, as well as ORC fungi, can have adaptations for aerial dispersal of spores^{13,14} and can grow independently of their host through saprophytic activity^{12,15-20}. We therefore expect EEM and ORC fungi to be better able to establish on islands prior to their hosts compared to AM fungi, and EEM and orchid plant species to be less impacted by the mycorrhizal filter than AM plant species²¹.

104. This statement about aerial dispersal merits references for ORC, if they exist.

We agree that this would be useful here but are unaware of any such references, so we were not able to add any.

178, 180. Can you mention how they "varied"?

Coefficients/ effect sizes of each analysis can be found in the Supplementary Information. We have opted not to focus on the specific effect size here as this is not a major result we wish to highlight in the main text. We have, however, added details of effect sizes of the main differences we highlight across mycorrhizal types in the results section.

(L 168-172)

Across oceanic islands globally, we found support for dispersal limitation of native angiosperm plant species that associate with AM fungi (Fig. 1, Fig. 2, SI Tables 1-6). Specifically, compared to mainland floras, we found that native island floras had approximately 5% lower proportion of AM than EEM (EEM:AM $p < 0.001$; SI Table 2a M1) and 10% lower AM than NM plant species (AM:NM $p < 0.001$; SI Table 4a M1).

(L 187-190)

Native island floras showed a lower proportion of orchid species compared to mainlands. Specifically, compared to mainland floras, native island floras had approximately 3% lower orchid to non-mycorrhizal plant species (ORC:M $p = 0.06$, SI Table 3a M1; and ORC:NM $p < 0.001$, SI Table 6a M1), consistent with the establishment limitation for orchids on islands.

(L 209-212)

Human-mediated plant naturalizations affected global plant biogeographical patterns influenced by mycorrhizal fungi (Fig. 2). In the naturalized flora, we found evidence of an increase of approximately 1% of the representation of EM relative to AM ($p < 0.001$, SI Table 2a M2) and approximately 10% AM relative to NM plant species on islands ($p < 0.001$, SI Table 4a M2).

260-264. Avoid repetitive unnecessary jargon, e.g. "ORC plant species" are "orchids".

Thank you. We have gone through the manuscript and changed all occurrences of ORC plant species to orchids after introducing them as such.

292-3. Avoid overstatement, this study is about colonization of islands, not the rest.

In our work, we investigated colonization patterns of native island plants based on plant mycorrhizal type (different types of mycorrhizal mutualisms differentially influence colonization of islands), patterns in island endemism/diversification based on plant mycorrhizal type (diversification), patterns of invasive island plants based on plant mycorrhizal type (plant invasion risks), and how environmental variables influence biogeography of plants across mycorrhizal types in both islands and mainlands (global plant biogeography). Nonetheless, we have reworded this sentence to focus on the main effects of the mycorrhizal filter on colonization of native island floras, along with patterns in diversification and plant invasion risks. This final sentence now reads as follows (L 298-300:

This work provides strong evidence that the major types of mycorrhizal fungi differentially influence plant colonization of islands, with subsequent effects on plant diversification and invasion risks.

303-4. Estimated how?

*Please see van Kleunen, M., P. Pyšek, W. Dawson, F. Essl, H. Kreft, J. Pergl, P. Weigelt, A. Stein, S. Dullinger, ... M. Winter. 2019. The Global Naturalized Alien Flora (Glo NAF) database. Ecology **100**:e02542 for a description of how these were estimated. In brief, a subjective completeness score was given to the dataset. For example, when a checklist was relatively short, it was deemed likely incomplete; when a checklist was extensive, it was deemed as nearly complete. The three options used were (1) likely very incomplete, (2) likely incomplete, and (3) likely nearly complete. We have opted not to include the details of estimation in our text as it is easily found in the above reference.*

323. This statement is incorrect and reference 43 is misinterpreted.

We have added an additional reference by (Vrålstad, 2004, Vrålstad 2000), considering the distinction between EM and ERM blurred through recent molecular work. We have clarified that this is the importance of the paper. This now reads as follows (L 236-332):

*We do not have a priori reason to expect differential colonization ability of ectomycorrhizal or ericoid mycorrhizal fungi because the distinction between these two groups is no longer clear. Specifically, molecular work has placed the model ericoid species *H. ericae* into a larger fungal complex containing EM fungi, with the suggestion that these two groups of mycorrhizal fungi may constitute a single guild^{49,50}. Therefore, we combine these mycorrhizal groups into one functional group in our analyses (EEM).*

395. This part about island age merits some discussion in the manuscript; why would age not play a role?

These ages are based on geological age and not biological age. Therefore, it may be that the lack of pattern we see with age is because this does not accurately reflect the time of plant colonization (exposed island), but the time since initial formation (with potential coverage with water and reemergence later in time). Overall, these ages likely overestimate time for colonization and the true age of the flora is hard to determine from these ages. We have added a note about age in the Supplementary Information/Methods section. This now reads as follows (L 347-348):

It is important to note that age reflects geological age, and not biological age, and so may not accurately reflect time from start of plant colonization.

REVIEWERS' COMMENTS:

Reviewer #1 (Remarks to the Author):

For the previous submission, I was impressed by the data and general conclusions, but had some reservations about the presentation, analysis, and interpretations. I have now read the author response to my comments and reviewed the revised manuscript.

I am happy to see that the authors took my comments to heart and addressed most of my concerns. I continue to think some folks will balk at the notion that ectomycorrhizal fungi can easily switch from saprotrophy to symbiosis, as a simple reading here might imply. This issue is really one of whether the hypotheses were properly motivated. Over all, I think the authors approach here is defensible.

Moreover, the paper is novel and I think will be of broad interest.

Reviewer #2 (Remarks to the Author):

>>The manuscript has improved. There are still misconceptions that need to be addressed.

From rebuttal: In contrast, other types of mycorrhizal fungi, including EEM, as well as ORC fungi, can have adaptations for aerial dispersal of spores^{13,14} and can grow independently of their host through saprophytic activity^{12,15-20}.

>>The fact that a few ectomycorrhizal fungi can grow in vitro on simple sugars does not lead to the gross generalization that ectomycorrhizal fungi can grow independently of their host through saprophytic activity! Most cannot do it. Please, you must be precise and accurate, if you mean a few species and in vitro, say it, otherwise you are inviting misinterpretation by non-expert readers. See Lindahl & Tunlid 2014, *New Phytologist*, Ectomycorrhizal fungi – potential organic matter decomposers, yet not saprotrophs.

From rebuttal: 104. This statement about aerial dispersal merits references for ORC, if they exist. We agree that this would be useful here but are unaware of any such references, so we were not able to add any.

>>If there are no references to support a statement, then don't make the statement or make it clear that it is speculation.

Reviewer #1 (Remarks to the Author):

For the previous submission, I was impressed by the data and general conclusions, but had some reservations about the presentation, analysis, and interpretations. I have now read the author response to my comments and reviewed the revised manuscript.

I am happy to see that the authors took my comments to heart and addressed most of my concerns. I continue to think some folks will balk at the notion that ectomycorrhizal fungi can easily switch from saprotrophy to symbiosis, as a simple reading here might imply. This issue is really one of whether the hypotheses were properly motivated. Over all, I think the authors approach here is defensible.

Moreover, the paper is novel and I think will be of broad interest.

Thank you for your time and effort with our manuscript.

Reviewer #2 (Remarks to the Author):

>>The manuscript has improved. There are still misconceptions that need to be addressed.

Thank you for your time reviewing our manuscript. Please find our responses below.

From rebuttal: In contrast, other types of mycorrhizal fungi, including EEM, as well as ORC fungi, can have adaptations for aerial dispersal of spores^{13,14} and can grow independently of their host through saprophytic activity^{12,15-20}.

>>The fact that a few ectomycorrhizal fungi can grow in vitro on simple sugars does not lead to the gross generalization that ectomycorrhizal fungi can grow independently of their host through saprophytic activity! Most cannot do it. Please, you must be precise and accurate, if you mean a few species and in vitro, say it, otherwise you are inviting misinterpretation by non-expert readers. See Lindahl & Tunlid 2014, New Phytologist, Ectomycorrhizal fungi – potential organic matter decomposers, yet not saprotrophs.

Thank you for this comment. We have included Lindahl & Tunlid 2014 as a reference questioning the ability of all EM to be free-living saprotrophs in order to include sources providing both work in support and work questioning this idea. In addition, we have modified this section to clarify that certain species within EM fungi have adaptations for aerial dispersal of spores and grow independently through saprotrophic activity. This section now reads as follows (L 84-90):

First, AM fungi lack adaptations for aerial dispersal (but see Chaudhary et al. 2020¹¹). Moreover, AM fungi cannot grow independently of their hosts¹². In contrast, individual fungal species of other types of mycorrhizae, particularly EEM, have adaptations for aerial dispersal of spores^{13,14} and can grow independently of their host through saprophytic activity^{12,15-20} (but see ²¹). We therefore expect EEM and ORC fungi to be better able to establish on islands prior to their hosts compared to AM fungi, and EEM and orchid plant species to be less impacted by the mycorrhizal filter than AM plant species²².

From rebuttal: 104. This statement about aerial dispersal merits references for ORC, if they exist. We agree that this would be useful here but are unaware of any such references, so we were not able to add any.

>>If there are no references to support a statement, then don't make the statement or make it clear that it is speculation.

We agree and have removed this mention of ORC aerial dispersal here. This now reads as follows noted in the previous response (L 84-90). We have also adapted the wording in the discussion about ORC dispersal and how this is linked to our findings. This now reads as follows (L 252-254):

These contrasting results may reflect the high ORC fungal specificity limiting island colonization overall, while the potential high dispersal ability of ORC fungi may contribute to orchid establishment on distant islands.